# Contribution of Fetal Inflammatory Response Syndrome (FIRS) with or without Maternal-Fetal Inflammation in The Placenta to Increased Risk of Respiratory and Other Complications in Preterm Neonates

**DOI:** 10.3390/biomedicines11020611

**Published:** 2023-02-18

**Authors:** Makoto Nomiyama, Takuya Nakagawa, Fumio Yamasaki, Nami Hisamoto, Natsumi Yamashita, Ayane Harai, Kanako Gondo, Masazumi Ikeda, Satoko Tsuda, Masato Ishimatsu, Yuko Oshima, Takeshi Ono, Yutaka Kozuma, Keisuke Tsumura

**Affiliations:** 1Department of Obstetrics and Gynecology, National Hospital Organization, Saga National Hospital, Saga 8498577, Japan; 2Department of Obstetrics and Gynecology, Faculty of Medicine, Saga University, Saga 8498501, Japan; 3Department of Pathology, Japan Community Health Care Organization, Saga Central Hospital, Saga 8498522, Japan

**Keywords:** bronchopulmonary dysplasia, chorioamnionitis, fetal inflammatory response syndrome, placenta, pregnancy, respiratory distress syndrome

## Abstract

This study classifies fetal inflammatory response syndrome (FIRS) based on the presence or absence of maternal-fetal inflammation in the placenta and clarifies the association of FIRS with neonatal morbidities. Women (330) who delivered at gestational ages of 22w0d-33w6d were enrolled and grouped into four based on FIRS and maternal/fetal inflammatory response (MIR/FIR) statuses: Group A: without FIRS and MIR/FIR (reference group); Group B: MIR/FIR alone; Group C: FIRS and MIR/FIR; and Group D: FIRS without MIR/FIR. The associations between bronchopulmonary dysplasia (BPD), adverse neonatal outcomes, extremely low gestational age and Groups B, C, and D were investigated after adjustment for potential confounders. Among patients with FIRS, 29% were in Group D. The risk of BPD was increased in Groups C (adjusted odds ratio (aOR): 3.36; 95% confidence interval (CI): 1.14–9.89) and D (aOR: 4.17; 95% CI: 1.03–16.9), as was the risk of adverse neonatal outcomes (Group C: aOR: 7.17; 95% CI: 2.56–20.1; Group D: aOR: 6.84; 95% CI: 1.85–25.2). The risk of extremely low gestational age was increased in Group D (aOR: 3.85; 95% CI: 1.56–9.52). Therefore, FIRS without MIR/FIR is not rare and may be associated with neonatal morbidities more than FIRS and MIR/FIR.

## 1. Introduction

Fetal inflammatory response syndrome (FIRS) is a state of fetal systemic inflammation associated with intra-amniotic inflammation. FIRS is associated with preterm birth and may cause prenatal damage to organs including the respiratory and central nervous systems, either alone or in conjunction with postnatal inflammation [1,2]. The diagnostic criterion for FIRS is an increase in the cord blood interleukin (IL)-6 concentration [3]. Among the histological acute inflammations of the placenta, extraplacental membranes, umbilical cord, funisitis, and chorionic vasculitis are termed a fetal inflammatory response (FIR) and can be used as diagnostic criteria for FIRS [4]. Early onset neonatal sepsis (EONS), bronchopulmonary dysplasia (BPD), intraventricular hemorrhage (IVH), periventricular leukomalacia (PVL), respiratory distress syndrome (RDS), and neonatal death are associated with FIRS [5,6,7,8,9,10,11,12]. However, whereas several studies focus on the use of FIR as a diagnostic criterion for FIRS, few reports regarding patients diagnosed with FIRS based on umbilical cord blood inflammation markers have been reported.

RDS and BPD are typical neonatal respiratory complications. A previous meta-analysis reported that the adjusted odds ratios of RDS and BPD are elevated in patients with FIRS [13]. Acute subchorionitis and chorioamnionitis of the placenta and extraplacental membranes, which are considered to be the basis of FIRS, are termed a maternal inflammatory response (MIR). However, the associations of BPD and RDS with MIR are unclear based on previous meta-analyses. The adjusted odds ratio of BPD increased or remained unchanged and that of RDS decreased or remained unchanged in patients with MIR [14,15]. Some patients with FIRS exhibit MIR/FIR, whereas others do not [3,16]. FIRS without MIR/FIR has been reported to be caused by sterile intrauterine inflammation [17], fetal anemia due to Rh-incompatible pregnancies [18], and maternal autoimmune diseases [19]. Microorganisms in the amniotic fluid and cytotoxic cytokines that cause intra-amniotic inflammation first cause local inflammation in the umbilical cord, respiratory tract, intestinal tract, and skin, during the development of FIRS with MIR/FIR. Microorganisms and cytokines in the amniotic fluid can reach the alveoli via the trachea and bronchi, and local inflammation in the alveoli progresses to systemic inflammation during the development of FIRS with MIR/FIR [20]. In patients with FIRS without MIR/FIR, no bacteria are detected in the amniotic fluid, and the cytokines in the amniotic fluid are not necessarily increased; therefore, the mechanism of the development of local lung damage remains unclear in these patients. The effects of local pulmonary inflammation and systemic hypercytokinemia on postnatal respiratory complications are not consistent, suggesting that the relationship between FIRS and neonatal respiratory complications may differ depending on the presence or absence of MIR/FIR. Additionally, the frequency of FIRS without MIR/FIR and its relationships with maternal background characteristics have not been investigated comprehensively.

Therefore, this study classifies FIRS according to the presence or absence of MIR/FIR and clarifies the association of FIRS with infant morbidities, with a focus on neonatal respiratory complications.

## 2. Materials and Methods

This single-institution retrospective cohort study was conducted with the approval of the institutional review board of National Hospital Organization Saga National hospital (approval number: R3–30).

Women pregnant with singletons who delivered at a gestational age of 22w0d to 33w6d between August 2014 and April 2021 were included in this study. Patients for whom the umbilical vein blood IL-6 concentration or placental histology was not available, those with an umbilical artery blood pH <7.10, and those with a major anomaly were excluded from the study. Patients who underwent a vacuum delivery and cases in which the fetus died were also excluded. Maternal steroids were administered in patients diagnosed with preterm premature rupture of the membranes (PPROM) at <34 weeks of gestation and those in whom a premature birth at <34 weeks of gestation was expected. Maternal antibiotics were administered prior to delivery in patients with PPROM or intra-amniotic inflammation and to prevent Group B streptococcus infections in newborns. Umbilical vein blood samples were obtained via venipuncture of clamped umbilical cords immediately after the delivery of the neonates and before the delivery of the placenta. The concentration of IL-6 in the umbilical vein blood was determined immediately by a chemiluminescence enzyme immunoassay (Human IL-6 CLEIA Fujirebio; Fujirebio, Tokyo, Japan). The detection range for IL-6 was 0.2–1000 pg/mL. At delivery, biopsies of the placenta, fetal membranes, and umbilical cord were fixed in 10% neutral buffered formalin.

FIRS was defined as an umbilical vein blood IL-6 concentration > 11.0 pg/mL. Staging of the MIR and FIR were performed according to Redline et al. and confirmed using the Amsterdam Placental Workshop Group Consensus Statement [21,22]. The patients were grouped based on their FIRS and MIR/FIR statuses: Group A: without FIRS and MIR/FIR; Group B: MIR/FIR alone; Group C: FIRS and MIR/FIR; and Group D: FIRS without MIR/FIR. Group A served as a reference group.

### 2.1. Diagnoses of Maternal Morbidities

PPROM was diagnosed as a collection of the amniotic fluid in the vagina on examination with a sterile speculum. Amniotic fluid leakage was confirmed using the insulin-like growth factor binding protein-1 test when the examination results were equivocal.

Pre-eclampsia (PE) was defined as de novo hypertension that developed at or after 20 weeks of gestation accompanied by one or more new-onset conditions at ≥20 weeks of gestation, including proteinuria, maternal organ dysfunctions (such as liver involvement, progressive kidney injury, stroke/neurological complications, and hematological complications), and uteroplacental dysfunction (such as fetal growth restriction, abnormal umbilical artery Doppler wave form, or stillbirth). In patients with preeclampsia, these symptoms normalize within 12 weeks after delivery [23].

Cervical insufficiency was defined as advanced cervical dilation and/or effacement in the absence of labor. Antepartum bleeding was defined as persistent or intermittent genital bleeding for at least seven days after 22 weeks of gestation.

### 2.2. Diagnoses of Placental Lesions

Tissue samples were obtained from the placenta (at least two samples), umbilical cord (typically three samples), and fetal membranes (at least one sample) at delivery. The samples were fixed in 10% neutral buffered formalin, embedded in paraffin, and sliced into 4-μm sections that were stained with hematoxylin and eosin (H&E) for microscopic assessment by a pathologist who was blinded to the patients’ clinical information. MIR was classified as Stage 1 (acute subchorionitis: patchy, diffuse accumulations of neutrophils in the subchorionic plate and/or membranous chorionic trophoblast layer), Stage 2 (acute chorioamnionitis: several scattered neutrophils in the chorionic plate or membranous chorionic connective tissue and/or the amnion), or Stage 3 (necrotizing chorioamnionitis: degenerating neutrophils, thickened, eosinophilic amniotic basement membrane, and at least focal amnionic epithelial degeneration). FIR was classified as Stage 1 (chorionic vasculitis/umbilical phlebitis: neutrophils in the wall of any chorionic plate vessel or the umbilical vein), Stage 2 (umbilical vasculitis: neutrophils in one or both umbilical arteries, with or without involvement of the umbilical vein), or Stage 3 (necrotizing funisitis or concentric umbilical perivasculitis: neutrophils, cellular debris, eosinophilic precipitates, and/or mineralization arranged in a concentric band, ring, or halo around one or more umbilical vessels). Pathologic features of the placenta that are indicative of maternal vascular malperfusion (MVM) include placental hypoplasia, infarction, retroplacental hemorrhage, distal villous hypoplasia, and accelerated villous maturation. Infarction hematoma was defined as a histologically confirmed hemorrhage encased by infarction [22]. According to Redline et al., diffuse chorioamniotic hemosiderosis was histologically defined as the diffuse deposition of retractile golden brown hemosiderin crystals in the chorioamniotic layers of the chorionic plate and/or membranes on the H&E stained sections [24].

### 2.3. Diagnoses of Neonatal Mortality and Morbidities

RDS was diagnosed based on the neonate’s clinical presentation of symptoms including tachypnea, nasal flaring, expiratory grunting, cyanosis, and intercostal, subxiphoid, and subcostal retractions and characteristic radiographic signs of low lung volume and a diffuse, reticulogranular ground glass appearance with air bronchograms [25]. The diagnosis of BPD was based on the National Institute of Child Health and Human Development’s 2001 definition [26]. Infants who required supplemental oxygen for at least 28 postnatal days were diagnosed with mild, moderate, or severe BPD, depending upon the extent of oxygen supplementation and other respiratory support. Neonates who were able to breathe room air at 36 weeks’ postmenstrual age or at discharge (whichever occurred first) were diagnosed with mild BPD. Moderate BPD was diagnosed in neonates who required <30% oxygen at 36 weeks or discharge, and severe BPD was diagnosed in neonates who required ≥30% oxygen or positive pressure ventilation at 36 weeks or discharge.

IVH was diagnosed using cranial ultrasonography or cranial magnetic resonance imaging and classified according to the system proposed by Papile et al. [27]. PVL was diagnosed via cranial ultrasonography or cranial magnetic resonance imaging performed at an age of at least two weeks. EONS was defined as culture-proven or clinical sepsis with an onset at <72 h of life. Patent ductus arteriosus (PDA) was defined as the persistence of an open ductus arteriosus after birth with clinical symptoms confirmed by echocardiography and treated with indomethacin or surgery. Neonatal mortality was defined as the death of the infant before discharge.

Adverse neonatal outcomes were defined as neonatal mortality or the presence of BPD, PVL, IVH, or EONS [5].

The primary study outcomes were the associations between the prevalence of RDS, BPD, and FIRS with or without MIR/FIR after adjustment for potential confounders. The secondary study outcomes were the associations between the prevalence of PDA, adverse neonatal outcomes, extremely low gestational age, and FIRS with or without MIR/FIR after adjustment for potential confounders.

### 2.4. Statistical Analyses

Continuous variables are presented as median (interquartile range (IQR), and categorical variables are presented as number (percentage). The clinical characteristics and short-term neonatal morbidity rates of the four patient groups were compared using the nonparametric Kruskal-Wallis test. Categorical variables were compared using the Fisher exact test. Bonferroni correction was used as a post-hoc test for categorical variables, and the Steel-Dwass test was used as a post-hoc test for continuous variables.

Logistic regression analysis was used to determine the associations between RDS, BPD, adverse neonatal outcomes, extremely preterm birth, and FIRS with or without MIR/FIR and MIR/FIR alone after adjusting for potential confounders. Groups A was used as a reference. Gestational age at delivery, maternal corticosteroid use, small for gestational age, and cesarean section were used to adjust for the analyses regarding RDS [28,29,30]. Gestational age at delivery and diffuse chorioamniotic hemosiderosis were included in the adjustment for the analyses regarding BPD [31,32]. The analyses regarding adverse neonatal outcomes were adjusted for gestational age at delivery, small for gestational age, diffuse chorioamniotic hemosiderosis, maternal corticosteroid use, and preeclampsia [5]. Diffuse chorioamniotic hemosiderosis, MVM, previous preterm birth, PPROM, and cervical insufficiency were adjusted for the analyses regarding extremely low gestational age [33]. The sample size required for a reliable multivariate logistic regression analysis was determined based on the assumption of 8–10 events per predictor variable [34,35]. At least 56 patients with RDS and 40 patients with BPD were required to satisfy this criterion.

Statistical significance was set at *p* < 0.05. All statistical analyses were conducted using EZR software (Saitama Medical Center, Jichi Medical University, Saitama, Japan), which is a graphical user interface for R software (The R Foundation for Statistical Computing, Vienna, Austria) [36].

## 3. Results

Among 421 singleton pregnancies that ended in preterm birth, 91 were excluded from this study (Figure 1). FIRS and MIR/FIR were concurrent in 19% (63/330) of the patients (Figure 2). FIRS without MIR/FIR was diagnosed in 8% (26/330) of the patients. MIR/FIR alone was diagnosed in 9% (31/330) of the patients. Neither FIRS nor MIR/FIR was detected in 64% (210/330) of the patients. The prevalence of FIRS with MIR/FIR and that of FIRS without MIR/FIR tended to decrease as the gestational weeks at delivery increased.

Patients with FIRS and MIR/FIR had a higher prevalence of PPROM than patients without FIRS and MIR/FIR (*p* < 0.001), with FIRS without MIR/FIR (*p* = 0.007) (Table 1). Patients with FIRS and MIR/FIR and with MIR/FIR alone had a lower prevalence of PE (*p* < 0.001, *p* = 0.04, respectively) and a lower prevalence of MVM (*p* < 0.001, *p* < 0.001, respectively) than patients without FIRS and MIR/FIR. The caesarean section rate was lower among patients with FIRS and MIR/FIR than among those without FIRS and MIR/FIR (*p* = 0.008).

Neonates with FIRS with or without MIR/FIR had a lower gestational age (*p* < 0.001, *p* = 0.02, respectively) and a higher prevalence of extremely low gestational age (*p* = 0.007, *p* < 0.001, respectively) and BPD (*p* < 0.001, *p* = 0.001, respectively), and adverse neonatal outcomes (*p* < 0.001, *p* < 0.001, respectively) than neonates without FIRS and MIR/FIR (Table 2). Neonates with FIRS and MIR/FIR had a higher prevalence of EONS than neonates without both FIRS and MIR/FIR (*p* < 0.001). Neonates with FIRS and MIR/FIR, with MIR/FIR alone had a lower prevalence of small for gestational age (*p* = 0.005, *p* = 0.03, respectively) than neonates without both FIRS and MIR/FIR. The umbilical vein IL-6 concentrations were highest in neonates with FIRS and MIR/FIR, followed by neonates with FIRS without MIR/FIR, those with MIR/FIR alone, and those without FIRS and MIR/FIR.

After adjusting for potential confounders, a diagnosis of FIRS and MIR/FIR (adjusted odds ratio (aOR): 3.36; 95% confidence interval (CI): 1.14–9.89) and a diagnosis of FIRS without MIR/FIR (aOR: 4.17; 95% CI: 1.03–16.9) increased the risk of BPD (Table 3). Diagnoses of FIRS and MIR/FIR decreased the risk of RDS (aOR: 0.25; 95% CI: 0.10–0.64), and a diagnosis of FIRS without MIR/FIR was not associated with RDS (Table 4). The risk of adverse neonatal outcomes was increased in patients with FIRS with MIR/FIR (aOR: 7.17; 95% CI: 2.56–20.1) and in those with FIRS without MIR/FIR (aOR: 6.84; 95% CI: 1.85–25.2) (Table 5). Patients with FIRS without MIR/FIR had an increased risk of extremely low gestational age (aOR: 3.85; 95 %CI: 1.56–9.52) (Table 6). The diagnoses of FIRS and MIR/FIR was not associated with extremely low gestational age.

## 4. Discussion

To the best of our knowledge, this is the first study to examine the relationship between FIRS and neonatal respiratory complications, neonatal morbidities, and maternal background characteristics in the presence or absence of MIR/FIR. FIRS in the absence of MIR/FIR occurred in 8% of preterm births up to 33 weeks, accounting for 29% of the patients with FIRS in this study. BPD and adverse neonatal outcomes were associated with FIRS with or without MIR/FIR. RDS was not associated with FIRS without MIR/FIR. Extremely low gestational age was associated with FIRS without MIR/FIR however, it was not associated with FIRS with MIR/FIR. The risk of PPROM was lower in patients with FIRS without MIR/FIR than in patients with FIRS with MIR/FIR, though it did not differ between patients with without FIRS and MIR/FIR and those with FIRS without MIR/FIR.

Several previous studies regarding FIRS without MIR, FIR, intra-amniotic infection or inflammation have been reported. It was first reported that elevated fetal blood IL-6 levels are associated with a shorter interval from cordocenteses to preterm birth and increased neonatal morbidity in PPROM or threatened preterm birth in 1998 [3]. This previous study also reported FIRS without MIR and intra-amniotic infection in 3% of the neonates, the interval from cordocenteses to preterm birth was short, and the infant morbidity was high. Another previous study reported that the frequency of FIR among patients with FIRS was only 56% [4]. An animal study conducted in 2009 reported that FIRS is caused by chronic hypoxia [37]. Fetal anemia in human Rh-incompatible pregnancies that develops into FIRS without intra-amniotic infection was first reported in 2011, highlighting the necessity to investigate whether non-infectious FIRS increases morbidity in infants [18]. In 2013, a prospective cohort study including 149 patients with PPROM reported that the frequency of FIRS was the highest when both MIR and microbial invasion of the amniotic cavity (MIAC) were observed. However, FIRS without MIR or MIAC was observed in 10% of the patients in the previous study, the interval from PPROM to delivery was short, and the association with neonatal morbidity was not investigated [17]. The frequency of brain imaging abnormalities in premature infants born to 103 patients with PPROM was 76% among patients with FIRS and MIR and 21% among patients with FIRS without MIR in 2016 [10]. A 2020 study reported that children born to pregnant women with autoimmune diseases had elevated cord blood levels of cytokines, though maternal hypercytokinemia and MIR were not observed [19]. Thus, previous reports on FIRS without MIR/FIR or intra-amniotic infection or inflammation have been limited to PPROM and threatened preterm birth in the maternal background, and the frequency of occurrence and characteristics of the maternal background are unknown. The association with neonatal morbidity has not been evaluated in numerous cases.

This retrospective cohort study included 330 preterm singletons born at 22–33 gestational weeks, regardless of the presence or absence of maternal complications. The presence or absence of FIRS and MIR/FIR, neonatal prognosis, and maternal background were analyzed. The frequency of FIRS without MIR/FIR was 8% of the total patient population and 29% of the total FIRS population, a relatively high frequency. This frequency is similar to the frequency reported in a study regarding patients with PPROM [17]. FIRS without MIR/FIR is associated with Rh-incompatible pregnancies and autoimmune diseases. However, no patients with FIRS without MIR/FIR in this study had Rh-incompatible pregnancies or autoimmune diseases. Patients with FIRS without MIR/FIR had a lower frequency of PPROM than those with FIRS and MIR/FIR. However, the maternal characteristics of patients without FIRS and MIR/FIR were not significantly different from those of patients with FIRS without MIR/FIR. Therefore, diagnosing of high-risk pregnancies with FIRS in the absence of MIR/FIR is challenging. It is crucial to recognize that FIRS is likely even in preterm birth without intra-amniotic infection or inflammation. We confirmed that neonates with FIRS without MIR/FIR have the same increased risk of BPD and neonatal adverse outcomes as neonate with FIRS and MIR/FIR. Therefore, even in preterm neonates born to mothers who are not at high risk of intra-amniotic inflammation, it is advisable to diagnose FIRS using cord blood and to take preventive measures against BPD and other morbidities in positive cases. Extremely low gestational age was associated only with FIRS without MIR/FIR. In the future, it may be possible to prevent extreme preterm births if the maternal background of FIRS without MIR/FIR can be characterized and treated. The association between FIRS with or without MIR/FIR and neonatal respiratory complications was interesting. BPD is associated with FIRS with and without MIR/FIR, but only neonates with FIRS and MIR/FIR have a lower incidence of RDS. The incidence of RDS is low in neonates with MIR/FIR only. This decrease in the incidence of RDS may be attributed to accelerated lung maturation due to local inflammation of the alveoli caused by pro-inflammatory cytokines and other factors in the amniotic fluid [20,38]. Conversely, BPD is associated with hypercytokinemia and does not necessarily involve local inflammation of the alveoli.

The effects of FIRS on the central nervous system were not examined in this study. In a previous study, patients with FIRS with MIR had a higher frequency of image-verified brain damage than those with FIRS without MIR. This may be due to the fact that cytokine levels in the umbilical cord blood are higher in patients with FIRS and MIR [10]. In this study, cord blood IL-6 levels in FIRS patients with MIR/FIR were higher than in FIRS patients without MIR/FIR; however, the presence or absence of MIR/FIR did not adversely affect BPD, neonatal outcomes, and FIRS patients without MIR/FIR were higher for incidence of extremely low gestational age. These findings suggest that the effects of FIRS on mothers and neonates may vary depending on the presence or absence of MIR/FIR and by differences in organs and cytokine levels in the fetal blood.

This study was not without limitations. First, this was a single-center, retrospective, cohort study, with a high number of excluded patients. Most of the excluded patients were cases in which the umbilical cord blood could not be collected. The umbilical cord blood IL-6 concentration is measured and placental histology is conducted for all premature deliveries at our institution to improve clinical management. Therefore, the retrospective nature of this study likely did not affect these parameters. Second, the definition of BPD used in this study was proposed in 2001, and the room air challenge test was not conducted. The frequency of BPD may have been affected, but the difference in the effect of FIRS with and without MIR/FIR on BPD was not affected. Third, the study period was seven years long; however, no significant changes in obstetric and neonatal management policies were identified during this period.

## 5. Conclusions

FIRS without MIR/FIR was not rare and was associated with neonatal morbidity as in FIRS with MIR/FIR and differed from FIRS with MIR/FIR in its association with maternal background, RDS, and extremely low gestational age. In the future, the effects of FIRS on organs other than the lung based on the presence or absence of MIR/FIR should be clarified. Additionally, differences in the maternal characteristics of patients with FIRS without MIR/FIR and those without FIRS and MIR/FIR must be investigated.

## Figures and Tables

**Figure 1 biomedicines-11-00611-f001:**
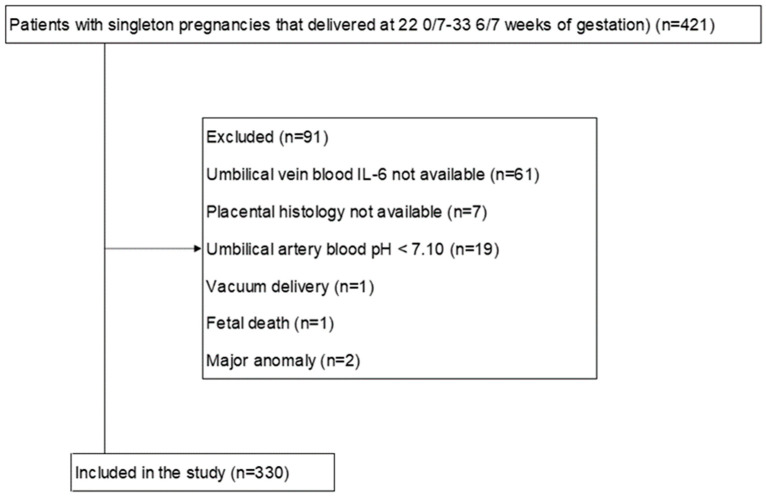
Patient flowchart.

**Figure 2 biomedicines-11-00611-f002:**
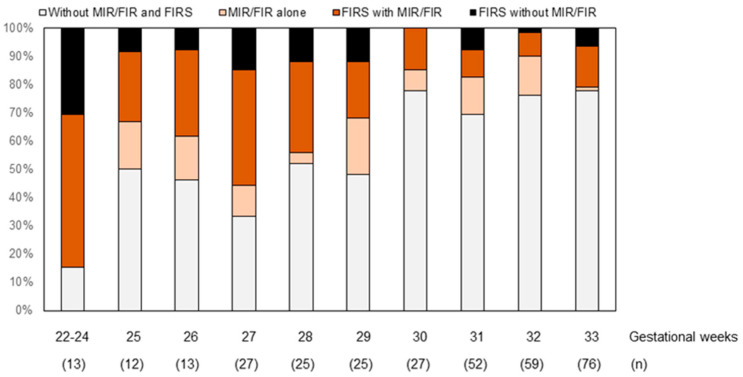
Incidence of FIRS with or without MIR/FIR according to gestational age. Abbreviations: FIRS, fetal inflammatory response syndrome; MIR, maternal inflammatory response; FIR, fetal inflammatory response.

**Table 1 biomedicines-11-00611-t001:** Maternal demographic and clinical characteristics according to the presence or absence of FIRS and MIR/FIR.

Factor	Without FIRS and MIR/FIR	MIR/FIR Alone	FIRS and MIR/FIR	FIRS without MIR/FIR	*p* Value	Adjusted *p* Value *					
	A	B	C	D		A vs. B	A vs. C	A vs. D	B vs. C	B vs. D	C vs. D
n	210	31	63	26							
Maternal age (years)	32.0 [28.0, 36.0]	31.0 [25.0, 34.0]	31.0 [28.5, 35.0]	29.5 [25.0, 33.8]	0.04	0.18	0.78	0.12	0.68	0.99	0.52
Primipara	86 (41.0)	13 (41.9)	27 (42.9)	15 (57.7)	0.45						
Previous preterm birth	29 (13.8)	5 (16.1)	8 (12.7)	4 (15.4)	0.97						
Maternal corticosteroid	147 (70.0)	20 (64.5)	48 (76.2)	17 (65.4)	0.6						
PPROM	57 (27.1)	15 (48.4)	36 (57.1)	5 (19.2)	<0.001	0.12	<0.001	1	1	0.17	0.007
Preeclampsia	52 (24.8)	0 (0.0)	0 (0.0)	3 (11.5)	<0.001	0.004	<0.001	1	1	0.53	0.14
Cervical insufficiency	14 (6.7)	5 (16.1)	8 (12.7)	5 (19.2)	0.07						
Antepartum bleeding	27 (12.9)	6 (19.4)	7 (11.1)	6 (23.1)	0.36						
Cesarean section	133 (63.3)	14 (45.2)	25 (39.7)	17 (5.4)	0.003	0.45	0.008	1	1	1	0.22
Maternal vascular malperfusion	64 (30.5)	0 (0.0)	1 (1.6)	5 (19.2)	<0.001	<0.001	<0.001	1	1	0.09	0.05
Diffuse chorioamniotic hemosiderosis	17 (8.1)	3 (9.7)	4 (6.3)	2 (7.7)	0.95						

* *p* values were adjusted using Bonferroni’s method. Data are presented as median (interquartile range) or number (percentage). Abbreviations: FIRS, fetal inflammatory response syndrome; MIR, maternal inflammatory response; FIR, fetal inflammatory response; PPROM, preterm premature rupture of the membranes; MVM, maternal vascular malperfusion; DCH, diffuse chorioamniotic hemosiderosis.

**Table 2 biomedicines-11-00611-t002:** Perinatal data according to the presence or absence of FIRS and MIR/FIR.

Factor	Without FIRS and MIR/FIR	MIR/FIR Alone	FIRS and MIR/FIR	FIRS without MIR/FIR	*p* Value	Adjusted *p* Value *					
	A	B	C	D		A vs. B	A vs. C	A vs. D	B vs. C	B vs. D	C vs. D
n	210	31	63	26							
Gestational age at delivery (weeks)	31.9 [30.0, 33.0]	31.1 [28.8, 32.2]	28.9 [27.1, 32.0]	28.8 [27.0, 31.8]	<0.001	0.06	<0.001	0.02	0.47	0.75	1
Extremely low gestational age	24 (11.4)	7 (22.6)	25 (39.7)	10 (38.5)	<0.001	0.54	0.007	<0.001	1	0.67	1
Male	120 (57.1)	19 (61.3)	33 (52.4)	19 (73.1)	0.33						
Birth weight (g)	1539.0 [1161.5, 1804.3]	1456.0 [1144.0, 1795.0]	1226.0 [920.5, 1580.0]	1052.0 [861.5, 1581.0]	0.001	0.99	0.009	0.01	0.2	0.11	0.86
Small for gestational age	63 (30.0)	2 (6.5)	6 (9.5)	8 (30.8)	0.001	0.03	0.005	1	1	0.19	0.13
Apgar score at 1 min	8.0 [5.0, 8.0]	8.0 [7.0, 8.0]	5.0 [3.0, 7.5]	3.5 [1.0, 7.8]	<0.001	0.84	<0.001	<0.001	0.004	0.003	0.34
Apgar score at 5 min	9.0 [8.0, 9.0]	9.0 [8.0, 9.0]	8.0 [7.0, 9.0]	7.5 [6.0, 9.0]	<0.001	1	<0.001	0.001	0.02	0.02	0.86
Umbilical artery pH	7.30 [7.26, 7.34]	7.33 [7.30, 7.36]	7.34 [7.29, 7.37]	7.29 [7.26, 7.34]	<0.001	0.02	0.003	0.98	1	0.13	0.11
Umbilical artery base excess	−3.0 [−5.0, −2.0]	−2.0 [−3.0, −1.5]	−3.0 [−5.0, −2.0]	−5.0 [−6.0, −4.0]	0.01	0.38	0.94	0.04	0.29	0.01	0.2
Umbilical vein blood IL−6 (pg/mL)	2.4 [1.6, 3.5]	4.1 [2.9, 6.3]	53.1 [24.7, 204.8]	23.6 [14.2, 32.7]	<0.001	<0.001	<0.001	<0.001	<0.001	<0.001	0.004
BPD	14 (6.7)	5 (16.1)	18 (28.6)	9 (34.6)	<0.001	0.47	<0.001	0.001	1	0.79	1
BPD (moderate−severe)	8 (3.8)	5 (16.1)	12 (19.0)	8 (30.8)	<0.001	0.09	0.001	<0.001	1	1	1
RDS	49 (23.3)	5 (16.1)	13 (20.6)	13 (50.0)	0.01	1	1	0.05	1	0.06	0.06
PDA	24 (11.4)	3 (9.7)	15 (23.8)	10 (38.5)	0.001	1	0.13	0.007	0.97	0.08	1
EONS	0 (0.0)	0 (0.0)	8 (12.7)	0 (0.0)	<0.001	1	<0.001	1	0.3	1	0.59
IVH	4 (1.9)	0 (0.0)	3 (4.8)	3 (11.5)	0.03	1	1	0.19	1	0.53	1
PVL	1 (0.5)	1 (3.2)	3 (4.8)	0 (0.0)	0.07						
Neonatal death	0 (0.0)	1 (3.2)	3 (4.8)	1 (3.8)	0.03	0.77	0.07	0.66	1	1	1
Adverse neonatal outcome	17 (8.1)	6 (19.4)	28 (44.4)	12 (46.2)	<0.001	0.55	<0.001	<0.001	0.13	0.27	1

* *p* values were adjusted using Bonferroni’s method. Data are presented as median (interquartile range) or number (percentage). Adverse neonatal outcomes include neonatal or infant death in the hospital, early-onset neonatal sepsis, intraventricular hemorrhage, bronchopulmonary dysplasia, and periventricular leukomalacia. Abbreviations: FIRS, fetal inflammatory response syndrome; MIR, maternal inflammatory response; FIR, fetal inflammatory response; IL-6, interleukin-6; RDS, respiratory distress syndrome; BPD, bronchopulmonary dysplasia; EONS, early-onset neonatal sepsis; IVH, intraventricular hemorrhage; PVL, periventricular leukomalacia; PDA, patent ductus arteriosus.

**Table 3 biomedicines-11-00611-t003:** Multivariate logistic regression analysis of the presence of bronchopulmonary dysplasia.

	Reference	Adjusted Odds Ratio	95% CI	*p* Value
MIR/FIR alone	Without FIRS and MIR/FIR	2.16	0.47–9.94	0.3
FIRS and MIR/FIR	Without FIRS and MIR/FIR	3.36	1.14–9.89	0.03
FIRS without MIR/FIR	Without FIRS and MIR/FIR	4.17	1.03–16.9	0.045
Gestational age at delivery	1 week	0.5	0.40–0.61	<0.001
Diffuse chorioamniotic hemosiderosis	no	13.7	3.39–55.2	<0.001

Abbreviations: FIRS, fetal inflammatory response syndrome; MIR, maternal inflammatory response; FIR, fetal inflammatory response; CI, confidence interval.

**Table 4 biomedicines-11-00611-t004:** Multivariate logistic regression analysis of the presence of respiratory distress syndrome.

	Reference	Adjusted Odds Ratio	95% CI	*p* Value
MIR/FIR alone	Without FIRS and MIR/FIR	0.29	0.09–0.95	0.04
FIRS and MIR/FIR	Without FIRS and MIR/FIR	0.25	0.10–0.64	0.004
FIRS without MIR/FIR	Without FIRS and MIR/FIR	1.49	0.48–4.62	0.49
Gestational age at delivery	1 week	0.61	0.53–0.70	<0.001
Maternal corticosteroid use	No use of maternal corticosteroids	0.25	0.13–0.49	<0.001
Preeclampsia	No preeclampsia	0.79	0.28–2.17	0.65
Small for gestational age	Not small for gestational age	0.76	0.32–1.80	0.53
Cesarean section	Vaginal delivery	1.33	0.67–2.64	0.42

Abbreviations: FIRS, fetal inflammatory response syndrome; MIR, maternal inflammatory response; FIR, fetal inflammatory response; CI, confidence interval.

**Table 5 biomedicines-11-00611-t005:** Multivariate logistic regression analysis of the presence of adverse neonatal outcomes.

	Reference	Adjusted Odds Ratio	95% CI	*p* Value
MIR/FIR alone	Without FIRS and MIR/FIR	1.98	0.50–7.81	0.33
FIRS and MIR/FIR	Without FIRS and MIR/FIR	7.17	2.56–20.1	<0.001
FIRS without MIR/FIR	Without FIRS and MIR/FIR	6.84	1.85–25.2	0.004
Gestational age at delivery	1 week	0.55	0.47–0.66	<0.001
Diffuse chorioamniotic hemosiderosis	No diffuse chorioamniotic hemosiderosis	9.57	2.52–36.4	<0.001
Maternal corticosteroid use	No maternal corticosteroid use	0.7	0.29–1.71	0.44
Preeclampsia	No preeclampsia	0.44	0.07–2.59	0.36
Small for gestational age	Not small for gestational age	1.19	0.38–3.77	0.77

Adverse neonatal outcomes include neonatal or infant death in the hospital, early-onset neonatal sepsis, intraventricular hemorrhage, bronchopulmonary dysplasia, and periventricular leukomalacia. Abbreviations: FIRS, fetal inflammatory response syndrome; MIR, maternal inflammatory response; FIR, fetal inflammatory response; CI, confidence interval.

**Table 6 biomedicines-11-00611-t006:** Multivariate logistic regression analysis of the presence of extremely low gestational age.

	Reference	Adjusted Odds Ratio	95% CI	*p* Value
MIR/FIR alone	Without FIRS and MIR/FIR	1.19	0.34–4.24	0.79
FIRS and MIR/FIR	Without FIRS and MIR/FIR	2.75	0.79–9.62	0.11
FIRS without MIR/FIR	Without FIRS and MIR/FIR	3.85	1.56–9.52	0.004
Diffuse chorioamniotic hemosiderosis	No diffuse chorioamniotic hemosiderosis	4.02	1.54–10.5	0.005
Maternal vascular malperfusion	No maternal vascular malperfusion	0.5	0.13–1.92	0.31
Previous preterm birth	No previous preterm births	0.33	0.09–1.21	0.1
Preterm premature rupture of the membranes	No preterm premature rupture of the membranes	0.88	0.39–1.99	0.76
Cervical insufficiency	No cervical insufficiency	1.43	0.45–4.59	0.55

Abbreviations: FIRS, fetal inflammatory response syndrome; MIR, maternal inflammatory response; FIR, fetal inflammatory response; CI, confidence interval.

## Data Availability

The datasets used in the current study are available from the corresponding author upon reasonable request.

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
