# Peer review of "Contribution of Fetal Inflammatory Response Syndrome (FIRS) with or without Maternal-Fetal Inflammation in The Placenta to Increased Risk of Respiratory and Other Complications in Preterm Neonates"

_biomedicines, 2023, doi:10.3390/biomedicines11020611_

Round 1

Reviewer 1 Report

I congratulate the authors on their excellent article entitled: Contribution of fetal inflammatory response syndrome (FIRS) with or without maternal-fetal inflammation in the placenta to increased risk of respiratory and other complications in preterm neonates.   I ask authors to perform minimal changes in their interesting manuscript:   line 282-283: replace "subjects" by "neonates" line 317 and line 318: replace "neonate" by "neonates" line 345: replace "is" by "was" (2x)  

Everything else is fine with this manuscript.

Author Response

Thank you for your review.

Response: I have changed: line 285-286: replace "subjects" by "neonates" line 322: replace "neonate" by "neonates" line 347: replace "is" by "was".

Reviewer 2 Report

Dear author, I just completed reading your Manuscript. You conducted a very interesting retrospective study. We know that cohort studies that are being conducted "as we speak", in regards to MIR/FIR with the use of IL-6 measurement will help us understand better the underlying pathology.

It is the first conducted study that examines the relationship of FIRS with or without MIR/FIR.
Your results point out that FIRS without MIR/FIR exists, something which was known, but it may be related with more severe neonatal morbidity in comparison to FIRS with MIR/FIR. That was what your study added, and it is well explained in the /Discussion/ section.

I have only minor comments to make 

Regarding the /Methods/ section:

In the section /"Diagnoses of maternal morbidities" please add the proper reference for each disease/ condition definition you mention.

In lines 146-157, please add the proper references regarding the used diagnostic criteria for RDS and BPD.

In the logistic regression model you conducted, you chose correctly the confounding factors for each dependent variable. Please add the proper references for each model, that justify the effect of the confounding factors to each disease/ condition (dependent variable).

The results are presented excellently, and the Tables are easy to read.

The /Discussion /and the /Conclusion /are consistent with the provided results

I really enjoyed reading your manuscript and I believe it will help those that are conducting research on the matter.

Author Response

Thank you for your review.

Response: In the section /"Diagnoses of maternal morbidities,  I added the referance [23] in line 117.

I add the references regarding the used diagnostic criteria for RDS [25] and BPD [26] in line 152,154.

I add the references for each model, RDS [28-30] BPD [31,32] adverse neonatal outcomes [5] extremely low gestational age [33] in line 188, 190, 192 and 195.